Giant worms chez moi! Hammerhead flatworms (Platyhelminthes, Geoplanidae, Bipalium spp., Diversibipalium spp.) in metropolitan France and overseas French territories

http://orcid.org/0000-0002-7155-4540 Justine Jean-Lou 1 justine@mnhn.fr
Winsor Leigh 2
Gey Delphine 3
Gros Pierre 4
Thévenot Jessica 5
1 Institut de Systématique, Évolution, Biodiversité (ISYEB), Muséum National d’Histoire Naturelle , Paris , France
2 College of Science and Engineering, James Cook University , Townsville, QLD , Australia
3 Service de Systématique Moléculaire, Muséum National d’Histoire Naturelle , Paris , France
4 Amateur Naturalist , Cagnes-sur-Mer , France
5 UMS Patrinat, Muséum National d’Histoire Naturelle , Paris , France
Riutort Marta
Electronic publication date: 2018 May 22
Publication date: 2018
Volume: 6
Electronic Location ID: e4672
Received 2017 Nov 16; Accepted 2018 Apr 6
Copyright: © 2018 Justine et al.
Copyright year: 2018
Copyright holder: Justine et al.
License: This is an open access article distributed under the terms of the Creative Commons Attribution License, which permits unrestricted use, distribution, reproduction and adaptation in any medium and for any purpose provided that it is properly attributed. For attribution, the original author(s), title, publication source (PeerJ) and either DOI or URL of the article must be cited.
License URL: https://creativecommons.org/licenses/by/4.0/

Keywords: Platyhelminthes, Barcoding, France, Land planarian, Overseas French territories, Citizen science, Invasive alien species

Funding: ATM Barcode ATM Origines This work was supported by ATM Barcode and ATM Origines from Muséum National d’Histoire Naturelle, Paris, France. The funders had no role in study design, data collection and analysis, decision to publish, or preparation of the manuscript.

==============================
Background

Species of the genera Bipalium and Diversibipalium, or bipaliines, are giants among land planarians (family Geoplanidae), reaching length of 1 m; they are also easily distinguished from other land flatworms by the characteristic hammer shape of their head. Bipaliines, which have their origin in warm parts of Asia, are invasive species, now widespread worldwide. However, the scientific literature is very scarce about the widespread repartition of these species, and their invasion in European countries has not been studied.

Methods

In this paper, on the basis of a four year survey based on citizen science, which yielded observations from 1999 to 2017 and a total of 111 records, we provide information about the five species present in Metropolitan France and French overseas territories. We also investigated the molecular variability of cytochrome-oxidase 1 (COI) sequences of specimens.

Results

Three species are reported from Metropolitan France: Bipalium kewense, Diversibipalium multilineatum, and an unnamed Diversibipalium ‘black’ species. We also report the presence of B. kewense from overseas territories, such as French Polynesia (Oceania), French Guiana (South America), the Caribbean French islands of Martinique, Guadeloupe, Saint Martin and Saint Barthélemy, and Montserrat (Central America), and La Réunion island (off South-East Africa). For B. vagum, observations include French Guiana, Guadeloupe, Martinique, Saint Barthélemy, Saint Martin, Montserrat, La Réunion, and Florida (USA). A probable new species, Diversibipalium sp. ‘blue,’ is reported from Mayotte Island (off South–East Africa). B. kewense, B. vagum and D. multilineatum each showed 0% variability in their COI sequences, whatever their origin, suggesting that the specimens are clonal, and that sexual reproduction is probably absent. COI barcoding was efficient in identifying species, with differences over 10% between species; this suggests that barcoding can be used in the future for identifying these invasive species. In Metropolitan south–west France, a small area located in the Department of Pyrénées-Atlantiques was found to be a hot-spot of bipaliine biodiversity and abundance for more than 20 years, probably because of the local mild weather.

Discussion

The present findings strongly suggest that the species present in Metropolitan France and overseas territories should be considered invasive alien species. Our numerous records in the open in Metropolitan France raise questions: as scientists, we were amazed that these long and brightly coloured worms could escape the attention of scientists and authorities in a European developed country for such a long time; improved awareness about land planarians is certainly necessary.

Introduction

Land planarians (Platyhelminthes, Geoplanidae) are predatory soil-associated animals. Although small species (generally less than 1 cm in length) such as Microplana spp. or Rhynchodemus spp. are autochthonous in Europe (Álvarez-Presas et al., 2012), large species are not. Reports of invasive alien flatworms in Europe in recent years (Sluys, 2016) include Arthurdendyus triangulatus from New Zealand, Platydemus manokwari originally from Papua New Guinea, Obama nungara from Brazil, and Parakontikia ventrolineata, Caenoplana coerulea, and Caenoplana bicolor from Australia (see Table 1 for authors of taxa and key references). All these species are conspicuous animals, several centimetres in length. Even larger are the species of Bipalium (and close genera), or ‘hammerhead flatworms’: these can be longer than 20 cm (Von Graff, 1899) and one species even attains a length of 1 m in elongated state (Kawakatsu, Makino & Shirasawa, 1982). In this paper, we focus on these giant species, and we report new findings obtained mainly by citizen science in metropolitan France and overseas French territories in the Caribbean (Guadeloupe, Martinique, and Saint Barthélemy), South America (French Guiana), and Indian Ocean (La Réunion, Mayotte). Five species were found, among which three can be attributed to known binomial taxa (Bipalium kewense, B. vagum, and Diversibipalium multilineatum) and two that are unnamed.

Table 1 Invasive land planarians found in Europe, authors of taxa and key references.

Taxon and authors	Synonyms	References for taxon	Main references for presence in Europe	
Arthurdendyus triangulatus (Dendy, 1894) Jones, 1999	Artioposthia triangulata	Dendy (1894), Jones (1999)	Boag et al. (1994)	
Platydemus manokwari De Beauchamp, 1963		De Beauchamp (1962)	Justine et al. (2014)	
Obama nungara Carbayo et al., 2016	Obama marmorata	Carbayo et al. (2016)	Carbayo et al. (2016)	
Parakontikia ventrolineata (Dendy, 1892) Winsor, 1991	Kontikia ventrolineata	Dendy (1892), Winsor (1991)	Álvarez-Presas et al. (2014)	
Caenoplana coerulea Moseley, 1877		Moseley (1877)	Álvarez-Presas et al. (2014), Breugelmans et al. (2012)	
Caenoplana bicolor (Von Graff, 1899) Winsor, 1991	Geoplana bicolor	Von Graff (1899), Winsor (1991)	Álvarez-Presas et al. (2014)	
Marionfyfea adventor Jones & Sluys, 2016		Jones & Sluys (2016)	Jones & Sluys (2016)	
Diversibipalium multilineatum (Makino & Shirasawa, 1983) Kubota & Kawakatsu, 2010	Bipalium multilineatum	Makino & Shirasawa (1983), Kubota & Kawakatsu (2010)	Mazza et al. (2016), This paper	
Bipalium kewense Moseley, 1878		Moseley (1878)	This paper	
Note:

This table provides complete information about authors and taxa and combination, thus making the general text lighter. Sluys (2016) listed additional species with limited records and information: Artioposthia exulans Dendy, 1901, Australoplana sanguinea (Moseley, 1877), Dolichoplana striata Moseley, 1877, Kontikia andersoni Jones, 1981.

Land planarians are dispersed between countries, and within countries, through the transport of plants. Winsor (1983a) summarized knowledge about the world distribution of B. kewense, listing the occurrence of the species in 39 territories; by 2004 the species was recorded in 45 territories (Winsor, Johns & Barker, 2004), and subsequently reported in Northern and Peninsular Italy, Sardinia, and Sicily (Gremigni, 2003); Czech Republic and Slovakia (Košel, 2002); Cuba (Morffe et al., 2016); Ecuador (Wizen, 2015); and Pakistan (M. Darley, 2003, personal communication to LW). As Sluys (2016) commented: ‘Almost every year B. kewense is found in new places: for example, this year (2016) it was found on São Miguel Island in the Azores and on São Tomé Island in the Gulf of Guinea.’ Although such reports from small remote islands are important for our knowledge of these invasive species (and we indeed add many new records of this type in this paper), we consider that the major finding of this paper is that several species of hammerhead flatworms are established in a European country, France, probably for more than 20 years. This highlights an unexpected blind spot of scientists and authorities facing an invasion by conspicuous large invasive animals.

The identification of land planarians from specimens or photographs is sometimes a futile exercise, in the absence of detailed anatomical study. In this paper, we tested identification with sequences of the cytochrome-oxidase 1 (COI). We confirm that barcoding with COI is efficient for the species studied here; in addition, our barcoding study revealed that all specimens in each species showed no genetic variability, suggesting that they are clonal, without sexual reproduction.

Material and Methods

Citizen science and collection of information

In 2013, one of us (JLJ) organized a citizen science network in France for collecting information about land planarians. This included a blog (http://bit.ly/Plathelminthe; Justine, 2017) and a Twitter account (https://twitter.com/Plathelminthe4). These efforts were advertised through the media (radio, television, and newspapers).

Reports of sighting of land planarians were received from citizens, mainly by email, sometimes by telephone. Photographs and details about locality were solicited, and only reports including this information were considered. Wrong records (slugs, myriapods, earthworms, leeches, caterpillars, nematomorphs, and nemerteans) were eliminated. Information collected from citizen science allowed monitoring of several land planarians (Justine, Thévenot & Winsor, 2014). Photographs were studied, and species were identified whenever possible. Only information relative to bipaliines is reported in this paper. Sometimes citizens provided records dating from before the survey, such as an amateur movie taken in 1999. Most citizens provided an authorization to use the photographs at the time of the initial contact by email. When we prepared this paper for publication, we sought authorization to use the photographs and to publish them under a Creative Commons Licence; only one of the citizens refused to provide the authorization, and some of them did not respond, probably simply because they changed their email addresses or did not check them. In these cases, we provide the scientific information about the presence of species, but we do not include the photograph of the worm or the name of the citizen in the paper.

Although these efforts were originally aimed at collecting information from Metropolitan France, they unexpectedly reached French territories in other continents and provided additional information and specimens.

Collection of specimens

In some cases, after examination of photographs, specimens were solicited from citizens who reported sightings; they were sent either alive or in ethanol by the citizens, registered in the collections of the Muséum National d’Histoire Naturelle, Paris (MNHN), and processed for molecular analysis.

When specimens were obtained alive, they were fixed in hot water then preserved in 95% ethanol. In some cases, some specimens were also fixed in hot water and preserved in 4% formaldehyde solution.

The species descriptions are based upon specimens and photographs obtained in this project.

Molecular sequences

For molecular analysis, a small piece of the body (1–3 mm3) was taken from the lateral edge of ethanol-fixed individuals. Genomic DNA was extracted using the QIAamp DNA Mini Kit (Qiagen, Hilden, Germany). Two sets of primers were used to amplify the COI gene. A fragment of 424 bp (designated in this text as ‘short sequence’) was amplified with the primers JB3 (=COI-ASmit1) (forward 5′-TTTTTTGGGCATCCTGAGGTTTAT-3′) and JB4.5 (=COI-ASmit2) (reverse 5′-TAAAGAAAGAACATAATGAAAATG-3′) (Bowles, Blair & McManus, 1995; Littlewood, Rohde & Clough, 1997). The PCR reaction was performed in 20 μl, containing 1 ng of DNA, 1× CoralLoad PCR buffer, 3 mM MgCl2, 66 μM of each dNTP, 0.15 μM of each primer, and 0.5 units of Taq DNA polymerase (Qiagen, Hilden, Germany). The amplification protocol was: 4′ at 94 °C, followed by 40 cycles of 94 °C for 30″, 48 °C for 40″, 72 °C for 50″, with a final extension at 72 °C for 7′. A fragment of 825 bp was amplified with the primers BarS (forward 5′-GTTATGCCTGTAATGATTG-3′) (Álvarez-Presas et al., 2011) and COIR (reverse 5′-CCWGTYARMCCHCCWAYAGTAAA-3′) (Lázaro et al., 2009), following Mateos et al. (2013). PCR products were purified and sequenced in both directions on a 3730xl DNA Analyzer 96-capillary sequencer (Applied Biosystems, Foster City, California, United States). Results of both analyses were concatenated to obtain a COI sequence of 909 bp in length (designated in this text as ‘long sequence’). Sequences were edited using CodonCode Aligner software (CodonCode Corporation, Dedham, MA, USA), compared to the GenBank database content using BLAST and deposited in GenBank under accession number MG655587–MG655618. For several specimens only ‘short’ sequences were obtained (Table 2).

Table 2 Specimens of bipaliines with molecular identification.

Species	MNHN	GenBank #	Date	Locality	Department/state	Country—continent	COI	Replicates	Collector	
BK	JL089	MG655587	12 November 2013	Saint Pée sur Nivelle	Pyrénées-Atlantiques	Met. France—Europe	Short	1	Consent not obtained	
BK	JL160	MG655605	23 May 2014	Cannes	Alpes-Maritimes	Met. France—Europe	Short	1	Iachia, Valeria	
BK	JL167	MG655615	24 August 2014	Orthez	Pyrénées-Atlantiques	Met. France—Europe	Short	1	Rougeux, Christian	
BK	JL174	MG655616	3 September 2014	Bassussary	Pyrénées-Atlantiques	Met. France—Europe	Long	1	Mercader, Elisabeth	
BK	JL176**	MG655617	5 September 2014	Auxerre (hothouse)	Yonne	Met. France—Europe	Long	1	Bellina, Arnaud	
BK	JL184	MG655603	October 2014	Ustaritz	Pyrénées-Atlantiques	Met. France—Europe	Short	1	Goyheneche, Iker	
BK	JL188	MG655604	8 October 2014	Miramar	Grande Porto	Portugal—Europe	Short	1	Soarès, Luciana	
BK	JL212	MG655592	19 December 2014	Mimbastes	Landes	Met. France—Europe	Long	1	Jouveau, Séverin	
BK	JL224	MG655607	23 February 2015	Trois Rivières	Guadeloupe	Guadeloupe—C. America	Long	1	Van Laere, Guy	
BK	JL233	MG655608	27 September 2014	Monaco	Monaco	Monaco—Europe	Long	3	Dusoulier, François	
BK	JL253	MG655609	21 March 2015	Trois Rivières	Guadeloupe	Guadeloupe—C. America	Short	1	Van Laere, Guy	
BK	JL254	MG655610	15 May 2015	Matoury	French Guiana	French Guiana—S. America	Short	2	Girault, Rémi	
BK	JL270	MG655594	23 April 2015	Ducos	Martinique	Martinique—C. America	Long	1	Lucas, Pierre-Damien	
BK	JL308	MG655602	8 September 2016	Morne Vert	Guadeloupe	Guadeloupe—C. America	Short	1	Coulis, Mathieu	
BV	JL073	MG655611	August 2013	Sanibel	Florida	USA—North America	Short	1	Justine, Jean-Lou	
BV	JL163	MG655613	July 2014	Sanibel	Florida	USA—North America	Short	1	Justine, Jean-Lou	
BV	JL164	MG655614	July 2014	Sanibel	Florida	USA—North America	Short	1	Justine, Jean-Lou	
BV	JL213	MG655593	29 November 2014	Anse-Bertrand	Guadeloupe	Guadeloupe—C. America	Long	1	Charles, Laurent	
BV	JL268	MG655595	December 2014	Montserrat	Montserrat	Montserrat—C. America	Short	1	Shoobs, Nathaniel F.	
BV	JL307	MG655601	19 November 2015	Morne Vert	Guadeloupe	Guadeloupe—C. America	Short	1	Coulis, Mathieu	
DM	JL177*	KT922162	30 September 2014	Léguevin	Haute-Garonne	Met. France—Europe	Long	1	Chaim, Florence	
DM	JL059	MG655618	15 June 2013	La Bastide de Serou	Ariège	Met. France—Europe	Short	1	Brugnara, Sébastien	
DM	JL142	MG655612	22 April 2014	Saubrigues	Landes	Met. France—Europe	Long	2	Robineau, Thiérry	
DM	JL161	MG655606	11 June 2015	Bellocq	Pyrénées-Atlantiques	Met. France—Europe	Long	1	Audiot, Marie-Claude	
DM	JL208	MG655589	11 June 2014	Bellocq	Pyrénées-Atlantiques	Met. France—Europe	Long	1	Audiot, Marie-Claude	
DM	JL209	MG655590	12 June 2014	Bellocq	Pyrénées-Atlantiques	Met. France—Europe	Long	1	Audiot, Marie-Claude	
DM	JL210	MG655591	June 2014	Bellocq	Pyrénées-Atlantiques	Met. France—Europe	Long	1	Audiot, Marie-Claude	
DM	JL298***	MG655600	1 June 2016	Novazzano	Ticino Canton	Switzerland—Europe	Long	1	Pollini, Lucia	
DBlue	JL280	MG655596	2015	Mtsamboro	Mayotte	Mayotte—Africa	Long	1	Charles, Laurent	
DBlue	JL281	MG655597	29 April 2015	Mtsamboro	Mayotte	Mayotte—Africa	Long	3	Charles, Laurent	
DBlue	JL282	MG655598	30 April 2015	Ouangani	Mayotte	Mayotte—Africa	Long	1	Charles, Laurent	
DBlue	JL284	MG655599	5 May 2015	Mtsamboro	Mayotte	Mayotte—Africa	Long	1	Charles, Laurent	
DBlack	JL090	MG655588	12 November 2013	Saint Pée sur Nivelle	Pyrénées-Atlantiques	Met. France—Europe	Short	1	Consent not obtained	
Notes:

Photographs are in Supplemental Information 1.

Most collectors were citizens; these collectors are professional: Arnaud Bellina, FREDON Bourgogne; Laurent Charles, Muséum Science et Nature, Bordeaux; Mathieu Coulis, CIRAD Martinique; Pierre-Damien Lucas, FREDON Martinique; Guy Van Laere, Parc National de Guadeloupe.

BK, Bipalium kewense; BV, Bipalium vagum; DM, Diversibipalium multilineatum; Dblue, Diversibipalium sp. ‘blue’; Dblack, Diversibipalium sp. ‘black.’

* JL177 already published (Mazza et al., 2016).

** Specimen from hot house, all others are from the open.

*** Specimen MCSN 719.990/77.590 kept in Museo Cantonale di Storia Naturale, Lugano, Switzerland, forwarded by Jean Mariaux (Geneva, Switzerland).

Trees and distances

MEGA7 (Kumar, Stecher & Tamura, 2016) was used to estimate genetic distances (kimura-2 parameter distance) and the evolutionary history was inferred from the kimura-2 parameter distance using the Neighbour-Joining method (Saitou & Nei, 1987); all codon positions were used with 1,000 bootstrap replications. The evolutionary history was also inferred using Maximum Likelihood (ML) method. The best evolutionary model for the data set was estimated in MEGA7 (Kumar, Stecher & Tamura, 2016) under the Bayesian Information Criterion (BIC) to be Hasegawa–Kishino–Yano model (Hasegawa, Kishino & Yano, 1985) with a discrete Gamma distribution and some sites invariables (HKY + G + I). The ML tree was computed in MEGA7 with 100 bootstrap replications.

A note about taxonomy of Diversibipalium

Morphology-based taxonomy of land planarians is based on a suite of characters, especially those afforded by internal anatomy, and in particular those of the reproductive system (Winsor, Johns & Yeates, 1998). Reproductive organs are only available in sexually mature specimens and require extensive histological preparations for their description. Unfortunately, many species of land planarians have been described from external morphology only. Some species only reproduce asexually (scissiparity) and thus do no show mature organs; this is especially the case of some invasive species when they are not in their region of origin. However, the bipaliines represent a special case because the external morphology, i.e., the presence of a ‘hammer’ head is distinctive of the subfamily, which thus can be easily differentiated if a photograph of the head is available. The genus Diversibipalium Kawakatsu et al., 2002 is a collective group created to temporarily accommodate species of the subfamily Bipaliinae whose anatomy of the copulatory apparatus is still unknown (Kawakatsu et al., 2002). For this reason, we attribute our two undescribed species, ‘black’ and ‘blue’ to this genus. We insist that attribution of species to the genus Diversibipalium does not mean that these species have characters in common—the only feature they share is our ignorance of their internal anatomy. These two species will be histologically examined and fully described by the authors elsewhere.

Results

Collection of information from citizen science

After the initial finding in June 2013 of two species of land planarians in his garden by Pierre Gros, an amateur entomologist and photographer, more than 600 reports were received over four years (June 2013–September 2017). Most records were from citizens, some from scientists or other professionals. Unexpectedly, these reports included mentions of more than eight species of land planarians (Justine, Thévenot & Winsor, 2014), the most recent being Marionfyfea adventor. Among these, 111 reports concerned bipaliines. Figure 1 is a map of these records in Metropolitan France.

Figure 1 Map of Metropolitan France (including Corsica) showing records of bipaliine flatworms.

Most records reported in this paper are outdoor but two are from hothouses. Note the concentration of records in the southern-east region, in the Department of Pyrénées-Atlantiques.

Results are presented here as follows: after an assessment of the identification of specimens from both morphology and molecules, separate paragraphs provide, for each species, a brief description and its range in Metropolitan France and overseas French territories, from both sampled specimens and photographs obtained through citizen science.

Molecular identification of sampled specimens

Sequences were obtained from specimens belonging to five species (Table 2), including three named species, B. kewense (specimens from 13 localities, 17 sequences including replicates), D. multilineatum (specimens from four localities, eight sequences including replicates), B. vagum (specimens from three localities, five sequences including replicates) and two unnamed species, Diversibipalium ‘black’ (one specimen from one locality, one sequence), and Diversibipalium ‘blue’ (specimens from two localities, six sequences including replicates).

A tree (Fig. 2) was constructed from an analysis of our new COI sequences and sequences from GenBank. Both NJ and ML trees showed comparable topologies, but the bootstrap values of branches, in both trees, were contrasted: 100% for all branches representing species, and very low for upper nodes. We thus considered that the trees were informative for showing the genetic identity of all specimens within a species, but not for inferring relationships between taxa. Thus, no further comment about interspecies relationships are given in the rest of this text; in that we follow the general principles of COI barcoding (Hebert & Gregory, 2005): ‘we emphasize that DNA barcodes do not aim to recover phylogenetic relationships; they seek instead to identify known species and to aid the discovery of new ones.’ We remarked, but do not comment, probable misidentification of certain sequences deposited in GenBank, such as Novibipalium venosum or the ‘D. multilineatum’ (HM346600).

Figure 2 Evolutionary relationships of taxa.

The tree shown was inferred using the Neighbour-Joining method. The percentage of replicate trees in which the associated taxa clustered together in the bootstrap test (1,000 replicates) are shown next to the branches, only when >70. The evolutionary history inferred by Maximum Likelihood method had similar topology. In both trees, branches representing the four species with several samples (Bipalium kewense, Bipalium vagum, Diversibipalium multilineatum, and Diversibipalium ‘Blue’) all had 100% bootstrap values, but bootstrap values for upper nodes were very low. We consider that the tree is informative for showing the genetic identity of all specimens within a species, but not for inferring relationships between taxa. New records with molecular information are indicated by *. For records in Metropolitan France, the number indicates the department code (i.e. 64: Pyrénées-Atlantiques).

Each of the three named species belonged to a clade with high (100%) bootstrap support (Fig. 2).

For B. kewense, the clade includes GenBank sequences from Spain, Azores Islands, and Cuba; our 13 new sequences (excluding replicates) are from seven localities in metropolitan France, three overseas French territories (Guadeloupe, Martinique, French Guiana), and two other countries, Monaco, and Portugal. All COI sequences were strictly identical.

For D. multilineatum, the clade includes GenBank sequences from Italy and France (sequence from specimen MNHN JL177, already published by Mazza et al., 2016), and our six new sequences (excluding replicates) are from three localities in metropolitan France. All COI sequences were strictly identical.

For B. vagum, no sequence was found in GenBank. Our five new sequences are from one overseas French territory (Guadeloupe) and two other countries, Montserrat (West Indies) and Florida, USA. All COI sequences were strictly identical.

For Diversibipalium ‘black’ from Metropolitan France and Diversibipalium ‘blue’ from Mayotte, each sequence was found to have no close match in GenBank sequences or our new sequences, suggesting that they each belong to a species which has never been sequenced for COI gene.

Distances between taxa

‘Short’ sequences were obtained from all specimens and ‘long’ sequences were obtained from only some of them. Distances between species of bipaliines were computed from two sets of sequences, ‘short’ sequences and ‘long’ sequences.

The first set included ‘short’ sequences and seven bipaliine taxa were available. Distances varied from 10.9% to 21.2% (Table 3). The closest taxa were B. kewense—D. multilineatum with an interspecific distance of 10.9%, and the most distant were Diversibipalium ‘blue’ and B. adventitium with 21.2%.

Table 3 Divergences between ‘short’ sequences of bipaliine flatworms.

	kewense	multilineatum	nobile	‘black’	‘blue’	vagum	
multilineatum	0.109						
nobile	0.131	0.131					
‘black’	0.149	0.164	0.163				
‘blue’	0.206	0.202	0.164	0.192			
vagum	0.140	0.168	0.163	0.140	0.159		
adventium	0.136	0.178	0.173	0.173	0.212	0.164	
Note:

There was a total of 266 positions in the final dataset.

The second set included only ‘long’ sequences and four bipaliine taxa were available. Distances were higher than with short sequences and varied from 15.9% to 25.9% (Table 4). The closest taxa were, again, B. kewense—D. multilineatum with an interspecific distance of 15.9%, and the most distant were Diversibipalium ‘blue’ and D. multilineatum with 25.9%.

Table 4 Divergences between ‘long’ sequences of bipaliine flatworms.

	kewense	multilineatum	‘blue’	
multilineatum	0.159			
‘blue’	0.230	0.259		
vagum	0.167	0.179	0.223	
Note:

There was a total of 857 positions in the final dataset.

Morphology, taxonomy, and distribution

Bipalium kewense Moseley, 1878

Morphology and colour pattern (Figs. 3–9)

Figure 3 B. kewense, general morphology.

Dorsal aspect of the planarian with a partial view of the ventral surface. Note the rounded posterior end indicating reproduction by scissiparity. Photo by Pierre Gros.

Figure 4 B. kewense, general morphology of the dorsal anterior end.

The expanded headplate, transverse black band (‘collar’) at the neck, and the median, paired lateral, and marginal dorsolateral dark longitudinal stripes are evident. Note that the median dorsal stripe does not pass onto the headplate. Photo by Pierre Gros.

Figure 5 B. kewense, side view of the headplate.

B. kewense hunts its earthworm prey using mechanoreceptors and chemoreceptors located along the leading margin of the headplate. These receptors are exposed when the papillae around the headplate are distended and moved like stubby fingers in an undulating motion to sense the environment, seen in this image. The under surface of the headplate is richly endowed with a variety of glands that include secretions with adhesive, lubricating and probably toxin-related functions. Photo by Pierre Gros.

Figure 6 B. kewense, general morphology, ventral surface.

The dark transverse neck band is incomplete ventrally, and the paired diffuse grey-purplish stripes delineate the off-white creeping sole. The position of the mouth is indicated by *, and the approximate position of the plicate protrusible pharynx within the body is evident as the pale area either side of the mouth. Photo by Pierre Gros.

Figure 7 B. kewense, predation on earthworm.

The flatworm initiates here the process of ‘capping’ the anterior end of the earthworm. Observed reactions of the prey suggest that it is at this stage that the planarian secretes a toxin to reduce prey mobility (Stokes et al., 2014). The planarian also produces secretions from its headplate and body that adhere it to the prey, despite often sudden violent movements of the latter during this stage of capture. Photo by Pierre Gros.

Figure 8 B. kewense, reproduction by scissiparity.

Some one to two days following feeding, the fission process is first manifested by a slight pinching of the body, some 1–2 cm. from the tail tip. Severance occurs when the tail tip adheres to the substratum and the rest of the planarian pulls away. Sexual reproduction outside their native habitat is restricted to individuals occupying outdoor situations in tropical or subtropical climates. Elsewhere they reproduce asexually. The links between sexuality and climate, and switching between scissiparity and egg cocoon production indicate that several interacting factors are involved, not least the availability of food and climatic variability (Winsor, Johns & Barker, 2004). Photo by Pierre Gros.

Figure 9 B. kewense, reproduction by scissiparity—the shed tail fragment.

The free tail fragment is immediately motile. It develops a head and pharynx within seven to 10 days, and within two to three weeks it is adult in form and behaviour (Connella & Stern, 1969). Asexual reproduction in B. kewense and some other land planarians is considered to underlie the colonizing success of these species (Hyman, 1951, p. 163). Photo by Pierre Gros.

Living specimens are long and thin and ranged in length from 100 to 270 mm (Table 5). Preserved specimens from which COI results were obtained, measured 170 mm (MNHN JL224), 120 mm (MNHN JL308), and 65 mm (MNHN JL270) in length, with the relative mouth: body length 41.2%, 41.7%, and 32.3%, respectively. None of the preserved specimens examined had a gonopore and thus they were considered to be non-sexual. The anterior end is expanded into a transversely semi-lunate-shaped headplate with recurved lappets (falciform). The dorsal ground colour is usually a light–mid ochre (Fig. 3), with five black to grey-coloured longitudinal stripes: a median, paired lateral, and paired marginal stripes which begin at or near the base of the headplate where it joins the body and the ‘neck.’ The dorsal headplate (Figs. 4 and 5) is usually the same colour as the body, or slightly darker, with recurved posterior margins. The median stripe is black, narrow, with sharp margins, extending caudally from below the neck over the entire body length, and is broadest over the pharyngeal area. Paired dark to pale brown coloured lateral stripes with diffuse margins, constant over the entire body length, are separated from the median and marginal stripes by an equal width of ground colour. The paired black, fine, marginal stripes, with sharp margins, extend the entire body length. The paired lateral and marginal stripes unite just behind the neck to form an incomplete black transverse neck band, interrupted dorsally by a small median gap, and ventrally by the creeping sole. The ventral headplate is a greyish colour with a light ochre margin. The ventral surface (Fig. 6) is a light ochre colour, with a distinct off-white creeping sole, delineated by paired, narrow, longitudinal diffuse grey–violet stripes beginning at the ventral termination of the collar, and extending the entire body length. In Fig. 7, we present evidence of predation on an unidentified native European earthworm, and in Figs. 8 and 9 evidence of reproduction by scissiparity where the shed fragment is immediately motile but does not possess the characteristic hammer-shaped head.

Table 5 Records of Bipalium kewense identified from photographs.

#	Date	Locality	Department/state	Country—continent	Origin of data	
K01	20 August 2017	Bora Bora	French Polynesia	French Polynesia—Oceania	Gerlach, Justin	
K02	13 October 2010	Basse-Terre	Guadeloupe	Guadeloupe—C. America	Guezennec, Pierre et Claudine	
K03	22 January 2014	Unknown	Guadeloupe	Guadeloupe—C. America	Consent not obtained	
K04	14 January 2007	Petit-Bourg	Guadeloupe	Guadeloupe—C. America	Lurel, Félix	
K05	19 February 2015	La Trinité	Martinique	Martinique—C. America	Delannoye, Régis	
K06	19 April 2016	Saint Joseph	Martinique	Martinique—C. America	Andrebe, Silvio	
K07	25 August 2017	Plaine des Cafres	La Réunion	La Réunion—Africa	Pronier, Pascal	
K08	3 November 2013	Cagnes-sur-Mer	Alpes-Maritimes	Met. France—Europe	Gros, Pierre	
K09	19 January 2014	Cagnes-sur-Mer	Alpes-Maritimes	Met. France—Europe	Gros, Pierre	
K10	5 November 2014	Cagnes-sur-Mer	Alpes-Maritimes	Met. France—Europe	Gros, Pierre	
K11	16 October 2013	Beaulieu-sur-Mer	Alpes-Maritimes	Met. France—Europe	Pelcer, Jean-Paul	
K12	21 July 2014	Nice	Alpes-Maritimes	Met. France—Europe	Gerriet, Olivier*	
K13	15 October 2014	Appietto	Corse-Sud (Corsica)	Met. France—Europe	Consent not obtained	
K14	17 October 2013	Pietrosella	Corse-Sud (Corsica)	Met. France—Europe	Senee, Patrick	
K15	23 August 2014	Arcachon	Gironde	Met. France—Europe	Consent not obtained	
K16	21 November 2002	Saint-Jean-de-Vedas	Hérault	Met. France—Europe	Peaucellier, Gérard	
K17	27 October 2014	Biscarosse	Landes	Met. France—Europe	Consent not obtained	
K18	27 September 2008	Hagetmau	Landes	Met. France—Europe	Jeannotin, Josette	
K19	22 September 2016	Nantes	Loire-Atlantique	Met. France—Europe	Consent not obtained	
K20	16 October 2014	Grimaud	Var	Met. France—Europe	Bernez, Alain	
K21	1 August 2014	Toulon	Var	Met. France—Europe	Consent not obtained	
K22	29 July 2014	Sens (Hothouse)	Yonne	Met. France—Europe	Burel, Jonathan**	
K23	23 October 2017	Peyrouse	Hautes-Pyrénées	Met. France—Europe	Tremosa, Clémence	
K24	17 December 2014	Arthez de Béarn	Pyrénées-Atlantiques	Met. France—Europe	Sillard, Dominique	
K25	17 September 2017	Billère	Pyrénées-Atlantiques	Met. France—Europe	Rolland, Geneviève	
K26	28 January 2018	Billère	Pyrénées-Atlantiques	Met. France—Europe	Rolland, Geneviève	
K27	20 September 2014	Bayonne	Pyrénées-Atlantiques	Met. France—Europe	Bonnefous, François	
K28	18 August 2014	Hasparren	Pyrénées-Atlantiques	Met. France—Europe	Voise, Mireille	
K29	22 April 2016	Jurançon (near)	Pyrénées-Atlantiques	Met. France—Europe	Pauchet, Marjolaine	
K30	29 April 2016	Nay	Pyrénées-Atlantiques	Met. France—Europe	Lamaille, Corinne	
K31	28 September 2014	Orthez	Pyrénées-Atlantiques	Met. France—Europe	Rougeux, Christian	
K32	22 August 2016	Saint Jean de Luz	Pyrénées-Atlantiques	Met. France—Europe	Centelles, Ruben	
K33	1 January 1999	Urcuit	Pyrénées-Atlantiques	Met. France—Europe	Esposito, Mario	
K34	14 September 2014	Urt	Pyrénées-Atlantiques	Met. France—Europe	Chanderot, Vincent	
K35	12 August 2017	Ustaritz	Pyrénées-Atlantiques	Met. France—Europe	Lescourret, Monique & Bernard	
K36	14 September 2014	Villefranque	Pyrénées-Atlantiques	Met. France—Europe	Consent not obtained	
Notes:

Photographs were obtained through citizen science; specimens were identified from photographs by the authors. No molecular identification was possible. There were 36 records (35 from outdoor and one from a hothouse). The name of the authors of photographs are indicated only when a formal consent to publish was obtained from the authors. Photographs are in Supplemental Information 2. For the first record, see also Gerlach (2017).

* Muséum d’Histoire Naturelle, Nice, France.

** FREDON Île de France.

Differentiation from other species

The specimens of B. kewense which were sent to us, or for which we received only photographs corresponded to published morphological descriptions of the species (Winsor, 1983a). B. kewense is differentiated externally from similar striped species by the incomplete black transverse band at the neck (the ‘collar’), the thin dorsal median longitudinal stripe that begins at or below the transverse neck band, the pattern and form of the dorsal and ventral stripes, and the relative position of body apertures (Winsor, 1983a).

Records obtained from citizen science

We obtained 50 records of B. kewense, including 14 confirmed by molecules (Table 2) and 36 from photographs only (Table 6). Localities where bipaliines were found in the open, generally in gardens, include Portugal (one record), Martinique (three), Guadeloupe (six), French Guiana (one), French Polynesia (one), La Réunion (one), Monaco (one), i.e., from seven territories in five continents (Europe, North America, South America, Africa, Oceania), and 36 from Metropolitan France (Fig. 1), from nine departments: Corse-Sud (Corsica) (two), Var (two), Gironde (one), Loire-Atlantique (one), Landes (three), Alpes-Maritimes (five), Yonne (two), Hautes-Pyrénées (one), and Pyrénées-Atlantiques (16). In addition, we received two reports in hothouses in the Department of Yonne. Among the 34 records in the open in Metropolitan France, 16, i.e., more than half, were from the department of Pyrénées-Atlantiques (Tables 2 and 6). The distribution of our records is shown in Fig. 1 for Metropolitan France (including Corsica). Dates of records ranged 1999–2017; the oldest record (1999) was in the Pyrénées-Atlantiques.

Table 6 Records of Diversibipalium multilineatum identified from photographs.

#	Date	Locality	Department/state	Country—continent	Origin	
M01	27 June 2010	Longages	Haute-Garonne	Met. France—Europe	Lombard, Yoann	
M02	22 March 2011	Longages	Haute-Garonne	Met. France—Europe	Lombard, Yoann	
M03	6 July 2016	Saint-Egrève	Isère	Met. France—Europe	Tuaillon, Jean-Louis	
M04	17 May 2017	Saint-Egrève	Isère	Met. France—Europe	Tuaillon, Jean-Louis	
M05	27 June 2016	Benquet	Landes	Met. France—Europe	Broustaut, François	
M06	28 March 2014	Cahors (Hothouse)	Lot	Met. France—Europe	Consent not obtained	
M07	4 July 2014	Andilly (Hothouse)	Val d’Oise	Met. France—Europe	Burel, Jonathan*	
M08	27 April 2015	Magny-en-Vexin	Val d’Oise	Met. France—Europe	Mellac, Céline	
M09	29 May 2016	Magny-en-Vexin	Val d’Oise	Met. France—Europe	Mellac, Céline	
M10	19 April 2010	Sames	Pyrénées-Atlantiques	Met. France—Europe	Grenier-Falaise, Nadine	
M11	7 April 2017	Billère	Pyrénées-Atlantiques	Met. France—Europe	Vincent, Jean-François	
Notes:

Photographs were obtained through citizen science; specimens were identified from photographs by the authors. No molecular identification was possible. There were 11 records, including two from hothouses. The name of the authors of photographs are indicated only when a formal consent to publish was obtained from the authors. Photographs are in Supplemental Information 2.

* FREDON Île de France.

Molecular results

The COI sequences were strictly identical for specimens from all localities where specimens were sequenced.

Diversibipalium multilineatum (Makino & Shirasawa, 1982)

Morphology and colour pattern (Figs. 10–14)

Figure 10 D. multilineatum, general morphology.

Dorsal aspect with a partial view of the ventral surface. The dark dorsal median stripe extends onto the headplate, and the headplate is more rounded than the falciform headplate of B. kewense. Note the rounded posterior end of the body indicating reproduction by scissiparity. Photo by Pierre Gros.

Figure 11 D. multilineatum, headplate.

On the headplate, the dark median dorsal stripe begins at the anterior third of the headplate and has a pronounced characteristic oblanceolate shape. Photo by Pierre Gros.

Figure 12 D. multilineatum, general morphology, anterior end.

The lateral dorsal stripes begin immediately behind the headplate. A transverse dark band (‘collar’) is absent. Photo by Pierre Gros.

Figure 13 D. multilineatum, ventral headplate morphology.

The fine, generally discontinuous mid ventral dark stripe extends from the anterior third of the headplate to the posterior end. There are also faint indications of the beginnings of the ventral paired lateral stripes on the headplate. Photo by Pierre Gros.

Figure 14 D. multilineatum, general morphology, ventral surface.

The three dark longitudinal stripes begin at the ‘neck’ and extend the length of the body. The position of the mouth is indicated by *, and the approximate position of the plicate protrusible pharynx within the body is evident by the diffuse line of the median stripe in this region. Photo by Pierre Gros.

Living specimens ranged in length from 150 mm (MNHN JL 177) to 210 mm (MNHN JL059). Representative preserved specimens from which COI results were obtained measured 85 mm (MNHN JL210), 65 mm (MNHN JL161A), and 60 mm (MNHN JL142A) in length (Table 5), with the relative mouth: body length 29.4%, 38.5%, and 41.7%, respectively. None of the specimens examined had a gonopore and thus they were considered to be non-sexual. The body is elongated (Fig. 10) with the anterior end expanded into a transversely semi-lunate-shaped headplate with rounded lappets (Figs. 11–13). Immediately behind the head the body narrows to form a ‘neck,’ then gradually broadens to the maximum width over the pharyngeal region, and tapers slightly to a rounded posterior end. The dorsal ground colour including the headplate is usually a light brown-ochre with five evenly spaced, black to dark brown longitudinal stripes: a median, paired lateral, and paired marginal stripes. The median stripe is black, and narrow with sharp margins. It has a pronounced characteristic lenticulate shape beginning at the anterior third of the headplate, then tapering to a thin dark stripe extending caudally along the entire body length, broadest over the pharyngeal area. Either side of the median stripe, each separated by an equal width of ground colour is a lateral stripe and submarginal stripe both of which join at the neck in the inner curvature of the headplate at the ‘neck’ and extend the entire body length. The lateral stripes are a black to dark brown colour with diffuse margins, approximately two- to three-times the width of the median stripe; the narrow, brown paired marginal stripes are approximately the same thickness as the median stripe. The ventral surface (Fig. 14) is a light brown ochre colour, generally slightly paler than that dorsally, with a distinct white creeping sole, delineated by paired, narrow, longitudinal brown stripes beginning at the ventral termination of the collar, and extending the entire body length. A finer, generally discontinuous mid-ventral dark stripe extends from the base of the headplate to the posterior end.

Differentiation from other species

The specimens of D. multilineatum which were sent to us, or for which we received only photographs corresponded to the published morphological description of the species (Makino & Shirasawa, 1983; Mazza et al., 2016). D. multilineatum is differentiated externally from similar striped species by the presence of the lenticulate-shaped beginning of the median stripe on the headplate, presence of distinct dark paired ventral median stripes, the thin, dark, generally incomplete mid-ventral longitudinal stripe, and the relative position of the mouth.

Records obtained from citizen science

We obtained a total of 19 records. One record was from Switzerland and 16 from outdoor locations in Metropolitan France, in the departments of Ariège (one), Haute-Garonne (three), Isère (two), Landes (two), Val d’Oise (two), and Pyrénées-Atlantiques (six); one record was confirmed two years in a row (2014–2015) in the same garden in Bellocq (Pyrénées-Atlantiques). In addition, two records were from hot-houses, in the Department of Lot (one) and Val d’Oise (one). Among the 16 records in the open in Metropolitan France, more than one third (six) are from the department of Pyrénées-Atlantiques. The distribution of our records is shown in Fig. 5 for Metropolitan France (including Corsica). Dates of records ranged 2010–2017; the oldest record (2010) was in the Pyrénées-Atlantiques (Tables 2 and 7).

Table 7 Records of B. vagum identified from photographs (no molecular identification).

#	Date	Locality	Department/state	Country—continent	Origin	
V01	21 June 2005	Cayenne	French Guiana	French Guiana—S. America	Girault, Rémi	
V02	15 May 2017	Macouria	French Guiana	French Guiana—S. America	Boutin, Élodie	
V03	12 May 2017	Saint-Laurent-du-Maroni	French Guiana	French Guiana—S. America	Muraine, François Xavier	
V04	26 July 2017	Saül	French Guiana	French Guiana—S. America	Sant, Sébastien	
V05	21 August 2017	Petit-Bourg	Guadeloupe	French Guiana—S. America	De Tienda, Marine	
V06	24 November 2013	Gosier	Guadeloupe	Guadeloupe—C. America	Consent not obtained	
V07	30 October 2016	Gosier	Guadeloupe	Guadeloupe—C. America	Brisson, Bernard	
V08	22 November 2013	Petit Bourg	Guadeloupe	Guadeloupe—C. America	Oettly, Olivier	
V09	22 November 2014	Petit Bourg	Guadeloupe	Guadeloupe—C. America	Marques, Maryvonne	
V10	29 April 2011	Petit-Bourg	Guadeloupe	Guadeloupe—C. America	Guezennec, Pierre et Claudine	
V11	21 October 2017	Petit-Canal	Guadeloupe	Guadeloupe—C. America	Charles, Laurent	
V12	29 November 2016	Le Moule	Guadeloupe	Guadeloupe—C. America	Consent non obtained	
V13	25 July 2010	La Trinité	Martinique	Martinique—C. America	Delannoye, Régis	
V14	18 November 2015	Morne Vert	Martinique	Martinique—C. America	Coulis, Mathieu	
V15	5 January 2018	Trois Ilets	Martinique	Martinique—C. America	Consent non obtained	
V16	1 April 2014	Saint Barthélemy	Saint Barthélemy	Saint Barthélemy—C. America	Moulard, Grégory	
V17	1 May 2014	Saint Barthélemy	Saint Barthélemy	Saint Barthélemy—C. America	Consent not obtained	
V18	11 May 2014	Saint Martin	Saint Martin	Saint Martin—C. America	Yokoyama, Mark	
V19	21 November 2015	Avirons	La Réunion	La Réunion—Africa	Consent not obtained	
V20	23 March 2017	Bras Panon	La Réunion	La Réunion—Africa	Saman-Latchimy, Teddy	
V21	29 March 2017	Le Tampon	La Réunion	La Réunion—Africa	Consent not obtained	
V22	26 October 2014	Petite Ile	La Réunion	La Réunion—Africa	Abonnenc, José	
V23	12 March 2016	Petite Ile	La Réunion	La Réunion—Africa	Le Gars, René	
V24	16 May 2014	Saint Louis	La Réunion	La Réunion—Africa	Faujour, Anne	
V25	8 April 2014	Saint Paul	La Réunion	La Réunion—Africa	Consent not obtained	
V26	16 March 2016	Saint Pierre	La Réunion	La Réunion—Africa	Collet, Jean	
V27	10 March 2013	Sainte Marie	La Réunion	La Réunion—Africa	Fontaine, Romuald	
V28	6 March 2016	Sainte Marie	La Réunion	La Réunion—Africa	Fontaine, Romuald	
V29	12 February 2009	unknown	La Réunion	La Réunion—Africa	Gilson, Michel	
V30	3 March 2010	unknown	La Réunion	La Réunion—Africa	Gilson, Michel	
V31	1 May 2011	unknown	La Réunion	La Réunion—Africa	Martiré, Dominique	
V32	28 October 2013	unknown	La Réunion	La Réunion—Africa	Martiré, Dominique	
V33	17 August 2015	unknown	La Réunion	La Réunion—Africa	Lacoste, Marie	
Note:

Photographs were obtained through citizen science; specimens were identified from photographs by the authors. No molecular identification was possible. There were 33 records, all from the outdoors. The name of the authors of photographs are indicated only when a formal consent to publish was obtained from the authors. Photographs are in Supplemental Information 2.

Molecular results

As for B. kewense, the COI sequences of D. multilineatum were strictly identical for specimens from all localities.

Bipalium vagum Jones & Sterrer, 2005

Morphology and colour pattern (Figs. 15–18)

Figure 15 B. vagum.

Specimen from French Guiana. The dorsal markings on this specimen are typical of the species. Note the dark patches on the headplate, continuous neckband, black median stripes, brown paired lateral stripes, and caudal black tip. Photo by Sébastien Sant, Parc Amazonien de Guyane.

Figure 16 B. vagum.

Specimen from Guadeloupe, West Indies. This specimen exhibits very light pigmentation, especially on the headplate, the indistinct brown paired lateral stripes and the caudal tip. Photo by Pierre and Claudine Guezennec.

Figure 17 B. vagum.

Specimen from Martinique, West Indies. In this specimen the headplate exhibits marked pigmentation so that it appears almost black. Photo Mathieu Coulis.

Figure 18 B. vagum.

Specimen from La Réunion, Indian Ocean. This specimen exhibits typical markings of the species. The paired dark patches on the headplate, and the dark pigmented caudal tip are clearly shown. Photo by Dominique Martiré.

Living specimens are medium sized, with one measuring around 36 mm (Table 5, observation V04, from a scaled photo). Preserved specimens, from which COI results were obtained, measured 27.5 mm (MNHN JL164), 25.6 mm (MNHN JL163), and 15 mm (MNHN JL307) in length, with the relative mouth: body length 60.7%, 50.4%, and 49%, respectively, and gonopore: body length 70.7% (MNHN JL163) and 72% (MNHN JL307).

Dorsal ground colour is a pale brown, with three black to brown dorsal longitudinal stripes: a median sharply demarcated broad black stripe, and two lateral dark brown stripes, less sharply delineated, all beginning at the transverse neck band, continuing the length of the full body, and often terminating in a well-defined black tip. The longitudinal stripes are separated from each other by an equal width of ground colour (Figs. 15–18).

Differentiation from other species

The specimens of B. vagum which were sent to us, or for which we received only photographs corresponded to the published morphological description of the species (Jones & Sterrer, 2005). B. vagum is distinguished externally from species of similar morphology by the combination of characters, especially its relatively small size, the transverse neck band that is continuous dorsally, from which the broad median black stripe originates, and the relative position of the body apertures.

Records obtained from citizen science

No record was obtained from Metropolitan France. We obtained 39 records (Tables 2 and 8), all in the open, from French Guiana (four records) and from five islands in the West Indies, including Montserrat (one) and four French territories, namely Guadeloupe (10), Martinique (three), Saint Barthélemy (two), and Saint Martin (one), and, from the Indian Ocean island of La Réunion (15); specimens from Florida, USA, were also sequenced. Unfortunately, in spite of the many photographic records from La Réunion, no specimen was received for sequencing, but the morphology and colour pattern were similar to other localities (Figs. 15–18). Dates of records ranged 2005–2017; the oldest record (2005) was from French Guiana (Tables 2 and 7).

Table 8 Records of Diversibipalium ‘blue’ identified from photographs (no molecular identification).

Date	Locality	Department/state	Country—continent	Origin	
7 March 2014	Unknown	Mayotte	Mayotte—Africa	Duperron, Benoît	
Note:

One record.

Molecular results

The COI sequences were strictly identical for specimens from all localities.

Diversibipalium sp. ‘black’ from Metropolitan France

Morphology and colour pattern (Figs. 19–21)

Figure 19 Diversibipalium sp. ‘black’ from Metropolitan France.

Drawings from photographs of three living specimens in dorsal view. The dorsal ground colour of the specimens is black, with no evidence of dorsal stripes. The scale (10 mm) is valid for the two specimens on the left, the specimen on the right has no scale.

Figure 20 Diversibipalium sp. ‘black’ from Metropolitan France, preserved specimen.

Specimen MNHN JL090. Dorsolateral aspect showing the partly protruded pharynx. Photo by Jean-Lou Justine.

Figure 21 Diversibipalium sp. ‘black’ from Metropolitan France, preserved specimen.

Specimen MNHN JL090. Ventral aspect. The ventral ground colour is grey, with the creeping sole a lighter tone. The pharynx is slightly protruded from the mouth, and the gonopore is evident as a small transverse white slit on the creeping sole some 2 mm below to the mouth. Scale is in mm. Photo by Jean-Lou Justine.

The living specimen attains a length of 20–25 mm. A preserved sexual specimen (MNHN JL090) is 20 mm long and 3.2 mm wide, with the mouth situated ventrally 6 mm (mouth: body length 30%), and gonopore 7.8 mm (gonopore: body length 39%) from the anterior end.

The dorsal ground colour of this small planarian is black, with no evidence of dorsal stripes (Figs. 19–21). The ventral surface is a light grey colour with paler creeping sole.

Differentiation from other species

In the absence of detailed data in the literature, it is difficult at present to determine whether Diversibipalium sp. 1 ‘black’ is a new species, or one of the small black species of Diversibipalium such as Diversibipalium sp. ‘Kuanmoto’ of Kawakatsu, Sluys & Ogren (2005).

Possible origin of this species

We do not propose any hypothesis concerning the geographic origin of this species, apart the fact that it is obviously not European, since no bipaliines are known from this continent.

Molecular results

The COI barcode of this specimen is clearly different from all other known sequences. We can safely claim that this species has never been sequenced before. Whether the species is already described or not is not an easy question to answer, and would require examination of mature specimens.

Diversibipalium sp. ‘blue’ from Mayotte (Indian Ocean)

Morphology and colour pattern (Figs. 22–26)

Figure 22 Diversibipalium sp. ‘blue’ from Mayotte, Indian Ocean, dorsal aspect.

The headplate of this small planarian is a brown colour, with a blue dorsum. This living specimen is approximately 45 mm long. Photo by Benoît Duperron.

Figure 23 Diversibipalium sp. ‘blue’ from Mayotte, Indian Ocean, dorsal aspect.

Specimen MNHN JL282. The headplate of this small planarian is a rusty-brown colour that extends to some irregular patches on the ‘neck.’ The dorsal ground colour is an iridescent blue–green (‘dark turquoise glitter’). Photo by Laurent Charles.

Figure 24 Diversibipalium sp. ‘blue’ from Mayotte, Indian Ocean, dorsal aspect.

Specimen MNHN JL282. Same specimen as in Fig. 25. Photo by Laurent Charles.

Figure 25 Diversibipalium sp. ‘blue’ from Mayotte, Indian Ocean.

Dorsal aspect of a regenerating specimen with a damaged anterior end. Specimen MNHN JL280. Under appropriate lighting, the colour of the specimen takes on a beautiful, almost metallic green colour. The iridescence and blue–green colour are lost on fixation, leaving the specimen a dark brown. Photo by Laurent Charles.

Figure 26 Diversibipalium sp. ‘blue’ from Mayotte, Indian Ocean.

Dorsal aspect of a regenerating specimen with a damaged anterior end. Specimen MNHN JL280. A small portion of the brown-pigmented ventral surface with the median pale creeping sole can be seen. Photo by Laurent Charles.

Unfortunately, scaled photos of this planarian are unavailable and the length of the living specimen could not be determined. The preserved sexual specimen is 9 mm long and 1 mm wide, with the mouth situated ventrally approximately 3.5 mm (mouth: body length 39%), and gonopore 6.5 mm (gonopore: body length 72.2%) from the anterior end.

The headplate in this beautiful, small planarian is a rusty-brown colour that extends to some irregular patches on the ‘neck.’ The dorsal ground colour is an iridescent blue-green (‘dark turquoise glitter’), and the ventral surface a very pale brown colour, with the creeping sole white to pale green. The iridescence and blue–green colour are lost on fixation, leaving a dark brown ground colour (Figs. 22–26).

Differentiation from other species

There are no other reports of a bipaliine planarian with this morphology.

Possible origin of this species

Mayotte and the Comoros are small volcanic islands which experienced intense human trade from centuries with the close islands and Madagascar and more distant territories including Asia. Any of these could be the origin of this species.

Records obtained from citizen science

We obtained records of this species only from Mayotte, from two independent observers, one who provided specimens and photographs and one who provided only photographs (Tables 2 and 9).

Table 9 Measurements of living specimens of bipaliines.

Species	MNHN specimen or photograph from Citizen Science	Locality	Body length (cm)	
Bipalium kewense	MNHN JL089	France	21	
	MNHN JL184	France	16	
	MNHN JL188	Portugal	25	
	MNHN JL224	Guadeloupe	21	
	MNHN JL270	Martinique	11	
	K04	Guadeloupe	13	
	K05	Martinique	20	
	K07	La Réunion	10	
	K24	France	20	
	K25	France	27	
	K28	France	15	
	K35	France	17	
Diversibipalium multilineatum	MNHN JL177	France	15	
	MNHN JL059	France	21	
Bipalium vagum	V04	French Guiana	3.6	
Note:

Measurements were estimated from photographs with scales obtained from citizen science (Supplemental Information 1 and 2).

Molecular results

The COI barcode of this specimen is clearly different from all other known sequences. We can safely claim that this species has never been sequenced before. Whether the species is already described or not is not an easy question to answer.

Discussion

Validity of COI for barcoding of bipaliine flatworms

Barcoding based on sequences of the mitochondrial gene COI has been proposed as a solution to the problem of species identification (Hebert et al., 2003). COI-based barcodes have been found to be effective in various groups, including butterflies (Lepidoptera) (Hebert et al., 2003) or fish (Ward et al., 2005). In flatworms (Platyhelminthes), although barcode based only on COI sequences might not be the best choice for some groups (Vanhove et al., 2013), recent studies showed that it efficiently differentiates species in groups such as monogeneans (Ayadi et al., 2017; Chaabane et al., 2016) and various triclads (Álvarez-Presas & Riutort, 2014) including land planarians (family Geoplanidae) (Álvarez-Presas et al., 2011, 2012, 2014).

The present study shows that COI short sequences, easily obtained from almost all specimens, have inter-specific distances of 10.9–21.2% (Table 3). These interspecific distances are high enough to differentiate species of bipaliines, especially in the absence of intra-species variation. Long sequences provide even higher inter-specific distances, ranging 15.9–25.9% (Table 4) but these are less easily obtained, and the database includes only four species. Of course, it might be objected that the current database (seven species with short sequences) is extremely limited in comparison to the number of species described in the bipaliines—more than 160 (Winsor, 1983a). However, the current database includes most invasive world-wide species, inter-specific distances are high, and intra-specific variation was almost inexistent for most species. For these reasons, we believe that identification of common invasive species of bipaliine flatworms can reliably be done from COI barcoding. Barcoding can be done from a very small worm, immature, or even a fragment. Moreover, COI barcoding can probably alert scientists to the presence of species not previously sequenced, if a sequence different from those reported in the present study is found.

The fact that some bipaliines do not reproduce sexually outside their native habitat or tropical and subtropical climates, but only by scissiparity (Winsor, 1983a), is probably one reason explaining why no variability was found in specimens, since specimens are clones, and no or very few mutations can happen. However, this explanation is not sufficient, since several populations from various origins, each cloning itself, could be present in the world. In contrast, for P. manokwari, COI sequences demonstrated the existence of at least two haplotypes in the world, probably corresponding to two populations and different ways of invasion of the world (Justine et al., 2015). Our current data on bipaliines suggest that one population is at the origin of the invasion for each species. This is particularly striking for B. kewense, with identical molecular records from several continents.

Persistence of B. kewense and D. multilineatum in the open in Metropolitan France

Bipalium kewense was originally described from specimens in one of the hot-houses in Kew, United Kingdom (Moseley, 1878). Originally from Vietnam to Kampuchea, the species is currently cosmopolitan (Winsor, 1983a). However, distinctions are important between a species which is found only in protected and restricted constructions such as hot-houses, and species which can freely live and reproduce in the open. Clearly, B. kewense is an invasive species in the open in countries with tropical moist or humid semitropical climates and appears to be restricted to anthropogenically-modified habitats; this is the case in the Caribbean, such as Guadeloupe or Martinique from where we obtained specimens. However, until recently (Justine et al., 2014), it was considered that B. kewense, in Europe, was only confined to hot-houses and thus not an invasive species. Examination of literature and citizen-science information (Fig. 1) now proves otherwise.

In France, the outdoors occurrence of B. kewense was reported in Orthez and Bayonne in 2005 (Vivant, 2005). Through citizen science, we obtained a movie of the worm filmed in the nearby locality of Urcuit in 1999. Moreover, we obtained information about the presence of the species in Arthez de Béarn, Hasparren, Villefranque, Urt (all in 2014), near Jurançon (2016), Nay (2016), and Saint Jean de Luz (2016), Billère and Ustaritz (2017) and, as in the report by Vivant (2005) in Bayonne and Orthez again (2014). We have obtained specimens from Saint-Pée-sur-Nivelle (2013), Ustaritz (2014), Bassussary (2014), and Orthez (2014). All these localities are in the Department of Pyrenées-Atlantiques, and we also have three records from the Department of Landes, north of Pyrenées-Atlantiques, along the Atlantic coast including Mimbastes (2014, with molecular information), Hagetmau (2008) and Biscarosse (2014) and one record from the Department of Hautes-Pyrénées, farther from the coast, in Peyrouse (2017) (Tables 2 and 6). The remark by Vivant that the animal was collected ‘five times in the last 20 years,’ the record from 1999, and the recent record and specimens in the same locality (Orthez) in 2014 strongly suggests that the species is now established in the open in Orthez and in several localities of the Department of Pyrenées-Atlantiques (Fig. 27). An alternative hypothesis would be that a single plant nursery near Bayonne acts as a continuing reservoir of planarians and that all these records are in fact specimens that escaped from recently bought plants, but which subsequently died after being released in the open; this hypothesis is falsified by records over several years in similar localities. Recently, one citizen in Billère (Pyrénées-Atlantiques) sent us repeated records in the same garden in September and December 2017 and January 2018, clearly showing numerous specimens alive outdoors, even in winter; they were found at various depths under the soil surface in January, clearly a way for the species to survive the cold season.

Figure 27 Map of the south-eastern part of France, showing numerous new bipaliine records.

Names of communes are indicated. Most records are from the Department of Pyrénées-Atlantiques, especially its lower part near the Atlantic Ocean.

We note that all our records are from gardens and that none were from places away from human presence; this can be expected from citizen science data.

We briefly comment on the climate of this region. The department of Pyrenées-Atlantiques is the most southern department on the Atlantic coast of France; it includes a mountainous region and a low altitude region along the ocean. The latter has an Atlantic climate. Within the department, we note that most records (Nay, Urcuit, Urt, Saint-Jean-de-Luz, Saint-Pée-sur-Nivelle, Ustaritz, and Bassussary) are from a small area around Bayonne, along the Atlantic coast (Fig. 27). The major limiting factor for a tropical species in Europe is, of course, low temperature. For a land planarian which is sensitive to drought and freezing, the numbers of days of drought in summer and the number of days of freezing temperature in winter are also important limiting factors. Detailed meteorological records are available for Biarritz, a locality close to Bayonne (Infoclimat, 2017): annual mean temperature is 13.7 °C, annual rain is 1,483 mm, even the dryer months (July and August) show a mean of 9–10 days with rain, and the number of days with temperature lower than −5 °C is only 1.5/year. This suggests that this region is particularly suitable for land planarians. Other localities in the south of France, such as Departments of Var and Alpes-Maritimes, and Corsica, both in Mediterranean climate, have higher temperatures and thus could be more suitable for tropical species, but they have longer periods of drought in summer (Infoclimat, 2017).

Interestingly, one record of D. multilineatum is also from the same department, in Bellocq (with records on two years), and the single record of Diversibipalium sp. ‘black’ is also from the same department, in Saint-Pée-sur-Nivelle, in a garden where B. kewense is also present. Other invasive land planarians found in the Pyrenées-Atlantiques include O. nungara, C. bicolor, and P. ventrolineata (data from citizen science). With a total of six species of invasive flatworm, clearly the Pyrenées-Atlantiques department is a hot spot of diversity and a small paradise for invasive land planarians.

For D. multilineatum, we have also two records in the same gardens in two consecutive years (Table 7). This suggests that this species also is established in the open in Metropolitan France, but the total number of records is lower (16 vs 34 for B. kewense). One of the records was of hundreds of animals.

A more detailed assessment of the ecoclimatic and other data for the distribution of invasive land planarians in France and French Territories is beyond the scope of this paper.

Do bipaliine land planarians qualify as invasive species in Metropolitan France?

We received several reports by citizens mentioning dozens of specimens in their gardens (Supplemental Information 1 and 2); in some cases, citizens repeatedly reported high numbers, even when worms were removed by hand and destroyed. Such reports justify the species as ‘invasive’ in the common, public sense of the word.

However, the term ‘invasive species’ has a more precise meaning in science. Invasive alien species (IAS) are defined by both the Convention on Biological Diversity and the International Union for Conservation of Nature as ‘species whose introduction and/or spread outside their natural past or present distribution threatens biological diversity’ (Convention on Biological Diversity, 2018; International Union for Conservation of Nature, 2018). Legal definitions are also available in various countries. In the USA, Executive Order 13112 (1999) defines an invasive species as ‘an alien species whose introduction does or is likely to cause economic or environmental harm or harm to human health.’ In Europe, the Institute for European Environmental Policy (Kettunen et al., 2009) uses the following definition: ‘Invasive alien species (IAS) are non-native species whose introduction and/or spread outside their natural past or present ranges poses a threat to biodiversity.’ The most recent legal text (European Parliament, 2014) reads (a few parts are deleted here for simplification): ‘(1) The appearance of alien species, whether of animals, plants, fungi or micro-organisms, in new locations is not always a cause for concern. However, a significant subset of alien species can become invasive and have serious adverse impact on biodiversity and related ecosystem services, as well as have other social and economic impact, which should be prevented. […] (2) IAS represent one of the main threats to biodiversity and related ecosystem services. […] (3) The threat to biodiversity and related ecosystem services that IAS pose takes different forms, including severe impacts on native species and the structure and functioning of ecosystems through the alteration of habitats, predation, competition, the transmission of diseases, the replacement of native species throughout a significant proportion of range and through genetic effects by hybridization.’

According to these definitions, bipaliines found in gardens in Metropolitan France and other localities mentioned in this paper should clearly be considered as IAS, because bipaliines are predators, and as such threaten the soil fauna. In absence of detailed ecological studies, we cannot estimate the exact impact of these worms on the fauna; the very large size of bipaliine flatworms, making them the largest terrestrial invertebrate predators, suggests that this impact is not negligible (Zaborski, 2002).

A precise classification of alien species based on their environmental impacts has recently been proposed (Blackburn et al., 2014); bipaliines fulfil three of the criteria listed in Table 1 of Blackburn et al. (2014): competition, predation, and poisoning/toxicity. The first two criteria are fulfilled by the predatory character of bipaliines, especially on larger prey (Ducey et al., 1999; Johri, 1952; Zaborski, 2002); the presence of tetrodotoxin (Stokes et al., 2014) fulfils the criterion of toxicity, and this is reinforced by reports of animals vomiting ingested bipaliines (Winsor, 1983b). However, in absence of ecological studies, bipaliines should currently be classified as ‘data deficient’ (Box 1 in Blackburn et al., 2014).

In conclusion, our results strongly suggests that bipaliines are IAS in Europe and the French overseas territories mentioned in this paper (Fig. 28), but an exact evaluation of their ecological impact requires ecological studies, which are outside the scope of this paper.

Figure 28 Map of the World, showing new records of bipaliine flatworms from French territories.

New records are from four continents (North America, South America, Polynesia, Africa).

How could 40 cm long invasive worms escape the attention of the scientists for 20 years?

At the beginning of our study, we were intrigued by the almost total absence of published information about the presence of bipaliines in France. The record by Vivant (2005) was the only one we could find, and since it was published in a rather obscure mycological journal, it certainly did not receive national nor international attention. Moreover, we are still amazed by the complete lack of response from scientific authorities at the presence of these worms. One of the early records we received (2013) was from a kindergarten in which the children were reportedly scared by hundreds of ‘small snakes’ on the grass (these were later identified as D. multilineatum). We also received a report of a citizen who showed a long hammerhead worm found on the fur of her cat to its veterinarian and was told it was a tapeworm (cestode). Other citizens explained that they tried to obtain identifications of land planarians from local universities and were told that the worms were leeches, and/or plain, uninteresting animals. Invasive land planarians were not known in France 10 years ago (Justine, Thévenot & Winsor, 2014) and the professionals involved in these anecdotes probably were never taught about them. Clearly, more education is needed about land planarians, which, in Europe, will be more and more often encountered by citizens and professionals in agriculture, landscaping, veterinary science, and medicine.

It is also amazing that the presence of such conspicuous animals never provoked a response from scientific authorities, although reports of tiny insect invasives often are followed by appropriate measures; again, the ignorance of professional scientists, science technicians, and amateur naturalists about land planarians was probably the reason. It is significant, in this respect, that the first recent mention of land planarians in France, by one of us (PG) was made public in an internet forum dedicated to insects. We expect that the measures taken at the European level will increase information about land planarians in the future (Tsiamis et al., 2016).

Conclusion

In this paper, we reported five species of bipaliine worms from Metropolitan France, a few European countries, and overseas French territories in three continents (Figs. 1, 27 and 28): much remains to be done, including a formal description of the two unnamed species. Of course, the results recorded here are only a very small part of the spread of these invasive species in the World. Initiatives like ours, including Citizen Science and molecular studies of selected specimens, should be undertaken worldwide. We have shown that molecular barcoding, based on COI, was efficient for the identification of the five species studied here, thus providing tools for future studies. We presented evidence that several species are spreading and that at least one of them is a predator of earthworms, which are important constituents of the soil fauna (Jones et al., 2001; Murchie & Gordon, 2013). We also demonstrated that bipaliines correspond well with the definition of ‘Invasive Alien Species’ in the European scientific (Kettunen et al., 2009) and legal (European Parliament, 2014) documents, but we recognize that a precise assessment of their impact on the local biodiversity is needed – but is outside the scope of this paper. Recently, a tendency to deny the risks posed by non-native species has emerged (Ricciardi & Ryan, 2018); in opposition to this ‘denialism,’ we strongly believe that invasive flatworms, as active predators, constitute a danger to native fauna wherever they are introduced.

Supplemental Information

Supplemental Information 1 Specimens of bipaliines with molecular data.

Photographs and details about bipaliines with molecular data, listed in Table 2.

Click here for additional data file.

Supplemental Information 2 Specimens from citizen science, without molecular data.

Photographs and details of specimens of bipaliines obtained from citizen science, without molecular data. Corresponds to Tables 5–8 of the paper.

Click here for additional data file.

Supplemental Information 3 French translation of the paper / Traduction française de l’article.

Click here for additional data file.

A complete French translation of the article is available as Supplemental File / Une traduction française intégrale de l’article est disponible comme fichier supplémentaire. Des vers géants chez moi ! Plathelminthes (Plathelminthes, Geoplanidae, Bipalium spp., Diversibipalium spp.) en France métropolitaine et dans les territoires français d’outre-mer. We thank all the citizens who participated in the survey; those who sent specimens are particularly thanked. Names of citizens, and sometimes scientists, who provided photographs and/or specimens are indicated in Tables 2, 5–8 and in the Supplemental Information. We apologize for not mentioning the names of citizens who kindly provided information but could not be contacted later for obtaining a formal consent. The support of various Fédérations Régionales de Défense contre les Organismes Nuisibles (FREDON), in Metropolitan France and overseas departments, is acknowledged. LW thanks Martin Darley for the specimen of B. kewense from Pakistan. Mrs. Vivant kindly provided a rare reference.

Additional Information and Declarations

Competing Interests

Author Contributions

DNA Deposition

Data Availability

Jean-Lou Justine is an Academic Editor for PeerJ.

Jean-Lou Justine conceived and designed the experiments, performed the experiments, analyzed the data, prepared figures and/or tables, authored or reviewed drafts of the paper, approved the final draft.

Leigh Winsor conceived and designed the experiments, performed the experiments, analyzed the data, prepared figures and/or tables, authored or reviewed drafts of the paper, approved the final draft.

Delphine Gey performed the experiments, contributed reagents/materials/analysis tools, authored or reviewed drafts of the paper, approved the final draft.

Pierre Gros performed the experiments, contributed reagents/materials/analysis tools, prepared figures and/or tables, authored or reviewed drafts of the paper, approved the final draft.

Jessica Thévenot performed the experiments, analyzed the data, prepared figures and/or tables, authored or reviewed drafts of the paper, approved the final draft.

The following information was supplied regarding the deposition of DNA sequences:

All new sequences have been uploaded to GenBank: under accession number: MG655587–MG655618.

The following information was supplied regarding data availability:

The raw data are included in Supplemental Files 1 and 2.

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
