# Peer review of "Giant worms chez moi! Hammerhead flatworms (Platyhelminthes, Geoplanidae, Bipalium spp., Diversibipalium spp.) in metropolitan France and overseas French territories"

_PeerJ, doi:10.7717/peerj.4672_

## Round 0.1 · original submission · Major Revisions

Two of the reviewers recommend major changes. After reading their comments, arguments and questions, I have found them pertinent and hence I ask you to read them carefully and try to follow their advise.
It is specially notorious their coincidence in certain aspects of their comments as:

1. At some points there is a lack of clarity on whether the morphology described refers to the animals collected for this work or it is a copy from original descriptions, this should be stated clearly. Moreover, one of the reviewers points to the need that in the present work the descriptions correspond to the animals collected to confirm or not their belonging to the species assigned.

2. Both also agree in their recommendation to remove the section on Predation of earthworms.

3. They show a certain concern on the affirmation that Bipaliinae in France are invasive. You should give a good definition of invasive species in one side, and in the other explain better in what evidences do you support your statement for Bipaliinae being invasive having into account that definition.

A part there are other comments from one or the other (in this sense I also find important the need to use better methodologies to infer a phylogeny) and from the second reviewer that finds only minor changes needed. However, this last criticises the title, something in which I do not agree, nonetheless you can also consider his opinion on this issue and consider making changes to the title.

·

Basic reporting

English is acceptable (but it is not my mother language). Literature is not suficiently and adequately explored. Article structure is deficient. Relevant results are hiddend by not original data.

Experimental design

Manuscript meets Aims and Scope of the Journal. Research question is well stated. Methods are not fully with sufficient detail.

Validity of the findings

Impact and novelty are obscured by many secondary informative information obtained from anecdotal observations and from literature, sometimes both sources are mixed and confounded with each other. Data are partially robust. Conclusion is not fully supported by the results.

Additional comments

In my enthusiasm for providing a quick review, I did not check my first revision for grammar errrors. Since my review is going to be published, I would ask you to accept the checked version of my review (below). If necessesary, I can send to you a version with corrections highlighted.

Best,
Fernando Carbayo.



1. 1. Basic reporting
English is acceptable (but it is not my mother language). Literature is not suficiently and adequately explored. Article structure is deficient. Relevant results are hiddend by not original data.
2. 2. Experimental design
Manuscript meets Aims and Scope of the Journal. Research question is well stated. Methods are not fully with sufficient detail.
3. 3. Validity of the findings
Impact and novelty are obscured by many secondary informative information obtained from anectodal observations and from literature, sometimes both sources are mixed and confounded with each other. Data are partially robust. Conclusion is not fully supported by the results.

In this manuscript the authors provide a number of new records of exotic bipaliine species introduced into France and overseas French territories. Part of the records were gathered with the collaboration of citizens during the last four years. Identification of the specimens from up to five species was conducted through examination of pictures of the specimens and COI gene fragment sequences, and compared against DNA sequences from GenBank. Since I am not expert in the molecular methods my comments on this point should be considered anecdotal.
Regarding the general aspects of the manuscript, I noticed main deficiencies as follows: (a) part of the headings and subheadings are out of place, (b) statements in Results are not based on evidences, (c) methods on which behavioural aspects (predation, reproduction) reported in Results and Discussion are not presented, and (d) behavioural aspects bring no novelty and are anecdotal. The whole manuscripts needs a profound revision and many sections should be deleted:

MATERIAL AND METHODS (M&M)
It is not stated how the descriptions of the species were done. It is not presented how behavioural and reproductive observations were conducted (although they should be deleted since they are out of the scope of the manuscript).

RESULTS
Description of B. kewense (lines 225-237) is a practically a copy-pasted version of Winsor’s (1983) description (as acknowledged by the authors). This is wrong. In this context, the purpose of the description is to provide evidence that the examined specimens do belong to kewense. Authors should have described their own specimens, and eventually conclude that the description matches that of B. kewense. The same is valid for the remaining known species.
“Predation of earthworms”, and “Morphological evidences of reproduction by scissiparity” of B. kewense (lines 242-249) are out of the scope of the work and Methods followed by the authors for obtaining associated data are not provided. These two subheadings should be deleted (out of scope).
Line 264: This statement has no support from the data and M&M (How did authors obtained these results? Did they offer non-earthworm prey?). These lines should be deleted.
After having read the description of B. kewense, I am not sure whether the description of external aspect of Bipalium multilineatum (lines 269-288) is original or it is a version of a description found elsewhere. Only descriptions based on collected specimens are valid for identification purposes. This should be stated unequivocally in M&M and the Results presented accordingly.
Lines 289-293 are not “Morphology” but comparative discussion and should be placed in a separate subheading. The same for every “Morphology” section of all species.
Source of the description of Bipalium vagum (lines 310-322) is uncertain. At least body dimensions are from the holotype found in Jones & Sterrer (2005) not from the specimens (or pictures) collected by the authors; this is wrong (see comment for B. kewense). It seems that the remaining description is based on the observation of the French specimens colelcted by the authors.
“Remarks” (lines 345-348; 363). Authors state that “The COI barcode of this specimen is clearly different from all other known sequences” and from this difference they conclude that it represents a different species. As presente, this is a weak argument. For instance, see the two terminals of B. venosum on Fig. 2; they are clearly different, but also considered to belong to just one species. The statement should be rewritten.

DISCUSSION
Supporting references should be given (lines 390-393).
Subheadings “Possible specific identity of the “black” and “blue” species”, and “A note about taxonomy of Diversibipalium” should be moved to the “Remarks” of either species.
Subheading “Predation in Bipaliines” (lines 408-431) should be deleted. The statements presented are purposeless and not derived from the M&M nor from Results.
Lines 438-444: It is concluded that “clearly, B. kewense is an invasive species”. Further in the same paragraph the authors declare to have examined the literature and citizen-science information to show that B. kewense is not confined to indoors but it may also be found outdoors. From having recorded B. kewense outdoors, authors conclude that this species is invasive. The meaning of is term, however, is not free from univocal meaning. What does it mean invasive for the authors? For a discussion, see Coulatti & MacIsaac (A neutral terminology to define ‘invasive’ species. Diversity and Distributions, (2004) 10, 135–141). Most records shown do no report the environs: indoors, outdoors, in natural habitats, natural forest areas, planted forests? There are indeed no reports on possible detrimental effects on the native fauna or rural activities (earthworm farms, grasslands...) apart from the two references provided (Winsor, 1983a, 1888b) in which it is reported that B. kewense is “a minor pest in earthworm farms”. To the best of my knowledge, this species has not been found in natural habitats outside its original distributional range area. The title of the manuscript is revealing in showing it: “flatworms [...] in metropolitan France [...]”. By reading the manuscript, I do not see reason for considering the species as invasive, independently from the habitats it is found. In the same rationale, the House Sparrow Passer domesticus could also be considered an invasive species. Author should make it clear what does ‘invasive’ mean, as suggested above.
It is unclear whether distributional data presented on lines 445-454 and Fig. 38 are the same shown in Fig. 1.

CONCLUSIONS
Some conclusions are not supported by the Results.
Last sentence is a belief (as they said) since no data are provided.

I would make additional minor comments to the manuscript, but at this stage, the text should best be rewritten. Nevertheless, I made minor suggestion in the PDF version of the manuscript. I recomend Major revision.


Fernando Carbayo

Reviewer 2 ·

Basic reporting

OK

Experimental design

OK

Validity of the findings

OK

Additional comments

General comments
This is a very interesting and well-presented study on a neglected group of invasive species. The figures are highly informative, particularly the photos of the specimens.
The professional English is generally good.
The use of citizen science is a very appropriate and original method to obtain detailed information on the distribution of these animals.
In addition to identification through external features the authors appropriately use DNA barcoding.

Specific comments
I have made annotations on the pdf of the manuscript, suggesting a rather limited number of changes in the text of the manuscript, as well as posing a few queries. I will not detail these remarks here, except the one concerning the use of French in the title of the manuscript.
I do strongly suggest that the authors strike the first four words (plus exclamation mark) from the title. The major reason is the use of the French ‘’Chez moi!’’. As I understand French, I can see that it is funny. However, most of the world population does not understand French and thus these words would not mean anything to the majority of the world population in general and of scientists in particular. Second, use of the word ‘’giant’’ to describe Hammerhead worms may be too much. Yes, for triclads they are big but usually people associate the word ‘’giant’’ with a different scale (elephants, blue whales). So, using ‘’giant’’ in the title may put people to stand on the wrong foot.

Annotated reviews are not available for download in order to protect the identity of reviewers who chose to remain anonymous.

·

Basic reporting

Overall, the article is clear and the English is correct. Relevant references are listed and the context is sufficient. Structure might be slightly improved. Results are interesting and useful for future studies.

Please check attached pdf for further comments.

Experimental design

The article contains original primary research including new reports of invasive hammerhead worms, new COI sequences and their analysis. Furthermore, two putative new Bipaliinae species found in France and Mayotte are reported.

Overall, the experimental design is correct.

Please check attached pdf for further comments.

Validity of the findings

Please check attached pdf for comments.

Additional comments

Please check attached pdf for comments.

---

## Round 0.2 · Minor Revisions

The reviewers now found that most of their suggestions have been followed, nonetheless there are still some minor changes proposed by the three of them, either on their comment or on an attached pdf of the ms. Please, try to follow their new recommendations (or justify why you do not do it). I hope to have your revised version soon, which I imagine I'll be able to check and accept without any further revision.

Best regards,

Marta

·

Basic reporting

This new version of the manuscript meets PeerJ standards with regard to organization of the the text. Unnecessary or out-of-scope sections (reproduction, predation) have properly been deleted. Species descriptions and comparison of the descriptions against those known species are robust. A reflections on the invasiveness of the species is properly stated. Authors are right in their criticism to the example I had provided on the invasiveness of House Sparrows in the first review. They also discuss what 'invasive' means and conclude that the species presented in the manuscript are at least potentially invasive. I made minor comments and corrections on the PDF version of the manuscript. I suggest publication with minor changes (see PDF archive).

Experimental design

Adequate.

Validity of the findings

Adequate. Data are robust. Conclusion is supported by the results.

Additional comments

See above.

Reviewer 2 ·

Basic reporting

OK

Experimental design

OK

Validity of the findings

OK

Additional comments

This revised version is a great improvement of the first draft of this manuscript, as the authors responded well to the detailed comments of the reviewers and completely restructured their manuscript.

However, I still feel that in their descriptions of B. kewense, B. multilineatum, and B. vagum they still fail to make fully clear whether the descriptions refer to the specimens collected (as it should) or represent a general description of the species. The confusion already starts with the first sentence of the description, which concludes that the animals correspond to the published morphological descriptions of the species. This conclusion should come at the end, i.e. after the description of the French animals. Generally, a (taxonomic) description is followed by a Discussion section in which it is explained to what extent the animals described correspond to a certain species and how they differ from other, closely related and/or similar-looking species. So, I suggest that the authors rename their heading ‘’Differentiation from other species’’ as ‘’Discussion’’ and start this with the first sentence of the current description of Morphology and Colour Pattern.

·

Basic reporting

The authors of the paper "Giant chez moi! Hammerhead flatworms (Platyhelminthes, Geoplanidae, Bipalium spp., Diversibipalium spp.) in metropolitan France and overseas French territories have satisfactorily answered the review comments and done many of the suggested improvements.

I only have very minor improvements to suggest:
> Line 73: Peninsular (instead of Peninsula).
> Line 181: I think that 'belong' should be 'belonging'.
> Line 462: winter (lower case).
> Line 205: one parenthesis is opened but not closed.

Experimental design

No comment.

Validity of the findings

No comment.

---

## Round 0.3 · accepted · Accept

Dear Jean-Lou,

I have revised the new version and I think it is ready to be accepted. I have only one little issue, as it was stated by the reviewer, in the description of species the comparison of the species with others is done under the title "discussion" in the taxonomic papers. So, it is not necessary to have the title "discussion and differentiation from other species", since discussion alone is understood as the place where this comparison is made. Please change that for the production phase.

Congratulations!

Marta

#